The development and application of pseudoviruses: assessment of SARS-CoV-2 pseudoviruses

Tan Conglian 1 2
Wang Nian 3
Deng Shanshan 2
Wu Xiaoheng 1 2
Yue Changwu 1
Jia Xu 2 jiaxu@cmc.edu.cn
Lyu Yuhong 1 yuhonglyu@126.com
1 Key Laboratory of Microbial Drugs Innovation and Transformation, Medical College, Yan’an University , Yan’an, Shaanxi , China
2 Non-coding RNA and Drug Discovery Key Laboratory of Sichuan Province, Chengdu Medical College , Chengdu, Sichuan , China
3 Chengdu Medical College , Chengdu, Sichuan , China
Wilke Claus
Electronic publication date: 2023 Dec 6
Publication date: 2023
Volume: 11
Electronic Location ID: e16234
Received 2023 Mar 21; Accepted 2023 Sep 14
Copyright: © 2023 Tan et al.
Copyright year: 2023
Copyright holder: Tan et al.
License: This is an open access article distributed under the terms of the Creative Commons Attribution License, which permits unrestricted use, distribution, reproduction and adaptation in any medium and for any purpose provided that it is properly attributed. For attribution, the original author(s), title, publication source (PeerJ) and either DOI or URL of the article must be cited.
License URL: https://creativecommons.org/licenses/by/4.0/

Keywords: SARS-CoV-2 pseudovirus packing system, Neutralizing antibody, Vaccine, SARS-CoV-2 variant, Drug, Virus-host interactions

Funding: National Natural Science Foundation of China 32170119 National Natural Science Foundation of China 31870135 This research was supported by the National Natural Science Foundation of China #1 under Grant (No. 32170119); and the National Natural Science Foundation of China #2 under Grant (No. 31870135). The funders had no role in study design, data collection and analysis, decision to publish, or preparation of the manuscript.

==============================
Although most Coronavirus disease (COVID-19) patients can recover fully, the disease remains a significant cause of morbidity and mortality. In addition to the consequences of acute infection, a proportion of the population experiences long-term adverse effects associated with SARS-CoV-2. Therefore, it is still critical to comprehend the virus’s characteristics and how it interacts with its host to develop effective drugs and vaccines against COVID-19. SARS-CoV-2 pseudovirus, a replication-deficient recombinant glycoprotein chimeric viral particle, enables investigations of highly pathogenic viruses to be conducted without the constraint of high-level biosafety facilities, considerably advancing virology and being extensively employed in the study of SARS-CoV-2. This review summarizes three methods of establishing SARS-CoV-2 pseudovirus and current knowledge in vaccine development, neutralizing antibody research, and antiviral drug screening, as well as recent progress in virus entry mechanism and susceptible cell screening. We also discuss the potential advantages and disadvantages.

Introduction

As of March 10, 2023, there were 759 million confirmed cases of COVID-19, of which 10–20% will experience mid- to long-term-post effects that are dominated by fatigue, dyspnea, cognitive impairment, and psychological effects (WHO Coronavirus (COVID-19) Dashboard, 2022), defined as Persistent Post-COVID Syndrome (PPCS) (Oronsky et al., 2023). The pathogen responsible for COVID-19 is SARS-CoV-2 (Lu et al., 2020), an enveloped virus with the structural proteins spike (S), envelope (E), membrane (M), and nucleocapsid (N) (Jackson et al., 2021). The S protein is the key to the virus infection (Chen, Liu & Guo, 2020). The S protein is composed of two functional subunits: S1 and S2. S1 contains the Receptor-Binding domain (RBD), which is responsible for the binding of the virus to angiotensin-converting enzyme (ACE2) on the host cell surface, and S2 promotes the fusion of virus outer membrane and cell membrane (Chan et al., 2020). Moreover, some protease cleavage sites on the S protein interact with cell membrane surface proteases, such as serine proteases (TMPRSS2), Cathepsins, etc., and promote virus invasion (Soleimani, 2020; Hoffmann et al., 2020) (The process of SARS-CoV-2 entering cells is shown in Fig. 1.). As a highly pathogenic virus, studies of SARS-CoV-2 must be conducted in the biosafety level (BSL) three laboratory, limiting the development of anti-viral research. To address this issue, pseudoviruses encapsulated with S protein are designed to replace the wild SARS-CoV-2, facilitating the study.

Figure 1 The process of SARS-CoV-2 entry into the cell.

There are two hypotheses for this process: (A) If TMPRS2 is not present on the cell membrane, SARS-CoV-2 enters the host cell by the endocytic pathway, forms phagocytic vesicles, cleaves of S2 by cathepsin L, and mediates membrane fusion, releasing viral RNA into the cytoplasm (Nadeem et al., 2020). (B) The S1 subunit of the S protein binds to ACE2 to trigger TMPRSS2 to cleave the S2 subunit, resulting in SARS-CoV-2 membrane fusion with the cell occurs and release of viral particles as intracellular (Jackson et al., 2021). Figure modified from Servier Medical Art (http://smart.servier.com/), CC BY 3.0.

The S protein encasing viral nucleic acids from other sources makes up the recombinant particle known as SARS-CoV-2 pseudovirus (Nie et al., 2020), whose surface proteins share a high degree of structural similarity with natural viral proteins. Thus, it still has immunogenicity and can efficiently mediate viral entry into host cells, genes within the pseudovirus are usually defective, but they can replicate in susceptible host cells only for one round, meaning a low level of pathogenicity (Li et al., 2018). Additionally, compared to live viruses, reporter genes are typically incorporated into pseudoviruses, making it considerably simpler to do quantitative and qualitative analysis. Therefore, SARS-CoV-2 pseudoviruses can replace highly pathogenic wild-type viruses in BSL2 laboratories, where they are widely used for drug screening and the development of vaccines. Additionally, in vitro investigations using a pseudovirus have also yielded information about the virus’s entry method into host cells.

In this review, we investigate the research on SARS-CoV-2 pseudovirus and describe its three construction methods and current applications. In particular, we focus on the methods to optimize the production and discoveries in COVID-19 vaccine monitoring, antibody and drug development, and host cell range. The purpose is to provide references for the application of SARS-CoV-2 pseudovirus in the study of COVID-19 pathogenesis and the search for the most effective prevention and treatment methods.

Survey methodology

A large number of documents (including clinical trials and reviews) on PubMed through the Internet were searched-which were then categorized and read meticulously. The key words are “SARS-CoV-2”, “SARS-CoV-2 pseudovirus”, “packaging system” and “application of pseudovirus”. The first aspect of the inclusion criteria is that the article has complete structure and sufficient materials, and the other is that it contains retrieval keywords.

Construction of SARS-CoV-2 pseudoviruses

Classification of SARS-CoV-2 pseudovirus packaging systems

The vesicular stomatitis virus (VSV) packaging system, the human immunodeficiency virus (HIV-1) lentivirus packaging system, and murine leukemia viral (MLV) retrovirus packaging system are the most extensively utilized packaging methods for SARS-CoV-2 pseudovirus. The main difference between them lies in the different sources of core genomic RNA.

The backbone of the VSV pseudovirus packaging system is provided by the replication-deficient VSV pseudovirus (G*ΔG-VSV), which contains a luciferase or fluorescent protein reporter motif but lacks the vesicular stomatitis virus glycoprotein (VSV-G) gene (Nie et al., 2020, p. 2). The construction of rVSV-SARS-CoV-2 pseudovirus is a very complex process, as shown in Fig. 2A, where recombinant poxvirus (Vtf7-3) infects juvenile hamster kidney (BHK-21) cells, expresses T7 RNA polymerase, which is used to direct the transcription of the full-length VSV genome from the T7-pVSV-reporter expression plasmid RNA and expresses four VSV proteins (N, P, G, L) (Pattnaik et al., 1992). In the presence of N, P, and L proteins, the T7-pVSV-reporter expression plasmid can replicate and amplify (Pattnaik & Wertz, 1990; Stillman, Rose & Whitt, 1995), and the G protein is expressed on the cell membrane and wrapped around the nucleocapsid to form an envelope with the outgrowth of pVSV-ΔG, which is named G*ΔG-rVSV (Schnell et al., 1996). The rVSV-SARS-CoV-2 pseudovirus can be obtained by infecting 293T cells that express S protein on the cell membrane after G*ΔG-rVSV transfection with the S expression plasmid (Schmidt et al., 2020). This method may leave some VSV residues, which can be removed by anti-VSV-G antibodies (Almahboub et al., 2020). 293T cells are co-transfected with two, three, or four plasmids to create the HIV-1 pseudovirus packaging system. As shown in Fig. 2B, 293T cells are co-transfected with an envelope plasmid expressing the S protein and an HIV-1 backbone plasmid lacking the gene encoding the HIV envelope glycoprotein env but containing a fluorescent reporter gene to form a dual plasmid system (Yang et al., 2020). The three-plasmid system divides the backbone plasmid into a packaging plasmid expressing gag and pol and a transfer plasmid containing the HIV reverse transcriptase gene, the cis-regulatory element required for integration, and the reporter gene (Crawford et al., 2020). A four-plasmid packaging system, further dividing the packaging plasmid into two plasmids which express gag-pol and rev, dramatically reduces the production of replicable virus and improves safety, but also loses a portion of the viral packaging titer (Dull et al., 1998). MLV-SARS-CoV-2-S pseudovirus is made by co-transfecting 293T cells with the MLV Gag-pol packaging vector, the MLV transfer vector expressing the reporter gene, and the S protein-encoding plasmid, and the packaging process, is shown in Fig. 2C (Giroglou et al., 2004).

Figure 2 Acquisition methods for SARS-CoV-2 pseudovirus based on three diûerent packaging systems.

(A) VSV packaging system. (B) HIV-1 packaging system. (C) MLV packaging system. Figure modified from Servier Medical Art (http://smart.servier.com/), CC BY 3.0.

Construction of relatively high titer SARS-CoV-2 pseudovirus

In general, the yield of SARS-CoV-2 pseudoviruses is lower than their actual requirements and can increase by modification of the S protein. Johnson et al. (2020) found that deletion of the last 19 amino acids of the cytoplasmic tail of the S protein, D614G and R682Q mutations all enhanced SARS-CoV-2 pseudotyped HIV-1 particle production and did not affect the neutralization ability against serum-neutralizing antibodies. Wang et al. (2023b) also reported that deletion of the furan site on the S protein gene increased the pseudovirus’s yield and infection efficiency. SARS-CoV-2 pseudotyped VSV particles with G614 S protein mutation have higher titer than the corresponding D614 pseudovirus (Korber et al., 2020).

Other pseudo-SARS-CoV-2 system

Besides the three packaging systems described above, researchers also constructed other pseudovirus systems. Construction of SARS-CoV-2 cDNA fragment in vitro, in which the N domain gene was replaced by GFP reporter gene, produced the RNA transcript of GFP/Δ N genome. In addition, the N domain gene was transduced by lentivirus to construct Caco-2 cells stably expressing the N gene (Caco-2-N). GFP/Δ N genome was injected into Caco-2-N cells to produce SARS-CoV-2-GFP/Δ N trVLP. Due to the lack of the N gene in the trVLP genome, only Caco-2-N cells can obtain virus replication and complete the life cycle of pseudo-SARS-CoV-2 (Ju et al., 2021; Zhang et al., 2022). Ma et al. (2022). constructed rSARS-CoV-2 using the split-virus-genome system. The full-length SARS-CoV-2 cDNA was divided into three parts: ORF1ab gene, S protein gene, and all the remaining structural genes carrying the reporter gene, and cloned into three plasmids respectively. These three plasmids were co-transfected with 293T cells to construct rSARS-CoV-2.

Sars-cov-2 pseudovirus for studying virus-host interactions

SARS-CoV-2 pseudovirus contains reporter genes that are easy to analyze qualitatively, and the membrane proteins can be altered by gene editing techniques so that investigating the role of these proteins in virus-host interactions can be studied.

SARS-CoV-2 pseudovirus employed to study the virus invasion mechanisms

The entry of SARS-CoV-2 pseudoviruses into cells can be easily observed by inserting a reporter gene into the S protein pellet. Ma et al. (2021) used a lentivirus packaging technology with a green cell membrane fluorescent probe, Dio, to label the viral envelope and a viral core genome carrying the red fluorescent protein mCherry to create a dual-color SARS-CoV-2 pseudovirus. mCherry and DiO co-localization signals indicated a single virus. Fluorescence imaging enables real-time tracking of cell infection by at the single particle level. mCherry and DiO separation shows that the virus core is released from the envelope into the host cell’s cytoplasm and is used to track how the pseudovirus penetrates four human respiratory cells. Their research unequivocally shows that endocytosis is the primary mechanism by which the pseudovirus enters host cells (Ma et al., 2021). The pseudoviruses and cells treated with gene editing technology or drugs can explore the mechanisms by which targets on host cells interact with SARS-CoV-2 to aid viral entry. The entrance of SARS-CoV-2 into 293T cells expressing hACE2 needed cathepsin L, according to research by Ou et al. (2020) using pseudovirus and cathepsin inhibitor-treated cells. Using gene editing cells and immunofluorescence labeling, Zhou et al. (2022) discovered that SARS-CoV-2 pseudovirus infection was linked to endocytosis, which uses Arf6-dependent CD147 endocytosis to enter host cell. Moon et al. (2023) used dual-reporter pseudovirus, HEK293T-ACE2, and HEK293T-ACE2-TMPRSS2 cells to verify that TMPRSS2-mediated membrane fusion plays a vital role in the SARS-CoV-2 infection process. Li et al. (2023) found that protein arginine methyltransferase 5 (PRMT5) plays a crucial role in infected cells. SARS-CoV-2 pseudovirus showed low infectivity towards their construct of PRMT5 knockdown A549 cells, and the infection rate of it was positively correlated with the amount of PRMT5 protein expressed by the host cells (Li et al., 2023).

SARS-CoV-2 pseudovirus for studying cellular susceptibility

SARS-CoV-2 pseudoviruses packaged with S proteins can mimic the process by which natural viruses bind to S protein cellular receptors to enter target cells (Ke et al., 2020). Ma et al. (2021) found the infection efficiency of four distinct respiratory cells to SARS-CoV-2 pseudovirus differed significantly, while VSV-G pseudovirus as a control group did not differ significantly. The reason was that SARS-CoV-2 infection effectiveness on host cells is closely related to the expression of ACE2 receptors on the cells, suggesting that the specific recognition of the receptor by the virus determines the susceptibility of different cells (Ma et al., 2021). Wang et al. (2022b) examined the ability of 27 mutant pseudoviruses to infect 20 Hela cells of animal origin, and they found that most mutations favor infection of domestic animals by the virus, and in particular, the T478I and N501Y mutations have been shown to infect chicken and mouse cells. The revelation that mutations in the S gene alter the host range and infectivity of SARS-CoV-2 provides a new perspective for the preventing and controlling COVID-19 outbreak (Wang et al., 2022a). Yi et al. (2020) applied pseudoviruses to study the virus’s susceptibility to brain cells. They found that SARS-CoV-2 readily entered ACE2-expressing neuronal somatic cells (Yi et al., 2020), suggesting that specific mature neural progenitor cells expressing ACE2 in the brain, such as choroid plexus epithelial cells, exhibit strong susceptibility to SARS-CoV-2 pseudovirus (Pellegrini et al., 2020). Chen et al. (2023b) observed that the pseudovirus could infect supporting and mesenchymal cells of hACE transgenic mice in vitro and cause severe pathological changes in the germinal tubules in mice in vivo.

Pseudovirus-based neutralization assay

The pseudovirus-based neutralization assay (PBNA) can simulate SARS-CoV-2 neutralization assays by constructing pseudoviruses with S protein. These pseudoviruses contain reporter genes that express a fluorescent signal only when they infect cells. The degree of signal reduction is positively correlated with the neutralizing antibody titer, and the neutralizing effect of the antibody can be determined by measuring the decrease in reporter gene expression. Nie et al. (2020) constructed pseudoviruses with VSV as the backbone to establish the PBNA, and validated the method in vaccine evaluation and antibody screening with good reproducibility and specificity. Tolah et al. (2021) demonstrated the method’s reliability for detecting neutralizing antibodies with PBNA compared to live virus micro-neutralization assays with a sensitivity of 85.94% and specificity of 100%. Thus, the PBNA method is easier to perform, more objective, sensitive, and more accurate than the live virus neutralization test assay. Because the SARS-CoV-2 variant pseudovirus is more readily available: it can be constructed by simply replacing the Spike protein expression plasmid (the Spike coding sequence of the variant), which has a substantial positive effect on the development of vaccines and antibody therapeutics.

SARS-CoV-2 pseudovirus employed to study the virus invasion mechanisms

The immunological effect of the COVID-19 vaccine is assessed by collecting vaccine immune sera for neutralization tests with SARS-CoV-2 to detect neutralizing antibody potency. The virus neutralization assay, which involves natural viruses, remains the gold standard for determining the immunological effects of COVID-19 vaccines. However, this method is not widely used in laboratory conditions, and the source of the virus limits it. Because of the advantages of more readily available and safer wide-type (WT) and variant pseudoviruses, collection of human or animal sera after vaccine immunization with SARS-CoV-2 pseudovirus for neutralization inhibition tests is more valuable to assess the antiviral efficacy of the vaccine. Table 1 summarizes the immune efficacy of four vaccines, BNT162b2 and Ad26. COV2-S, mRNA-1273, and CoronaVac against multiple SARS-CoV-2 variations. In general, the emergence of variations, especially omicron, reduces the neutralizing efficacy of the vaccines. Researchers have proposed booster vaccination in response to the rapidly mutating viruses and are working to develop new vaccines. Barros-Martins et al. (2021) applied this method to detect that heterologous ChAd/BNT was effective in inducing higher titers of neutralizing antibodies against B.1.1.7, P.1, and B.1.351 variant pseudoviruses compared to the ChAd/ChAd homologous group; Mahasirimongkol et al. (2022) detected neutralizing activity of antibodies against WT, Alpha, and Beta, Delta mutant pseudoviruses in sera following CoronaVac/ChAd heterologous inoculation. Jin et al. (2023) found that the heterologous aerosolized Ad5-nCoV improved the neutralization efficacy against BA.4/5 pseudovirus compared to three-does CoronaVac homologous boosting. PBNA against SARS-CoV-2 WT and variations are widely used to evaluate the viral neutralization efficacy of improved and new vaccines. For example, the rVSV vaccine expressing Delta-S protein and influenza matrix protein (M2e) (Ao et al., 2022), bacterial vector vaccine carrying S protein receptor binding domain gene (Zhou et al., 2023), and multi-epitope vaccine equipped with S and non-S surface protein targets (Wang et al., 2023a). The improvement method of vaccine is generally to find the Optimizing adjuvant. The immune sera of vaccine with Aluminum salts (Alum)-3M-052 as adjuvant showed higher antibody neutralization efficiency against both B.1. 617.2 and B.2. 12.1 pseudoviruses (Huang et al., 2023), and the RBD subunit vaccine assisted by QS21 + 3-O-desacyl-4′-monophosphoryl lipid A obtained stronger the anti-Delta and WT pseudovirus infection ability (Shi et al., 2023). It is also necessary to test the efficacy of COVID-19 vaccine in special populations. PBNA showed that vaccination can obtain anti-SARS-CoV-2 neutralizing antibody for special populations, such as patients with kidney failure, gastrointestinal cancer receiving systemic anti-cancer therapy and inborn errors of immunity, but it is usually weaker than ordinary people, so it is necessary to strengthen immunity (Lau et al., 2022; Erra et al., 2023; Ling et al., 2023).

Table 1 Using PBNA to detect the influence of SARS-CoV-2 variants of concern (VOCs) on the effectiveness of four vaccines.

Vaccine	WHO EUL holder	Sample size	Days post second vaccine dose	Reference strain	B.1.1.7	P.1	B.1.351	B.1.617.2	B.1.1.529	References	
BNT162b2	Pfizer/BioNTech	32	19–32d	WT	NA	NA	6.7-fold decreases	2.2-fold decreases	22.8-fold decreases	Muik et al. (2022)	
26	26d	D614G	1.3-fold decreases	NA	3.2-fold decreases	2.2-fold decreases	28.6-fold decreases	Evans et al. (2022)	
30	7–32d	WT	2.1-fold decreases	6.7-fold decreases	34.5-fold decreases	NA	NA	Tang et al. (2022)	
Ad26.COV2-S	Janssen	20	30d	WT	NA	NA	12.6-fold increases	5.97-fold increases	12.6-fold increases	Liu et al. (2022)	
mRNA-1273	Moderna	7	14d	D614G	NA	NA	8.9-fold decreases	NA	35.1-fold decreases	Wu et al. (2021a)	
8	7d	D614G	1.2-fold decreases	3.2-fold decreases	6.9-fold decreases	2.7-fold decreases	NA	Barouch et al. (2021), p. 19	
CoronaVac	Sinovac	20	14d	WT	2.9-fold increases	4.3-fold increases	5.5-fold increases	3.4-fold increases	12.5-fold increases	Wang et al. (2022a)	
Note:

WT, wild type (Wuhan strain); D614G, virus Spike protein acquired the D614G mutation, early in the pandemic, replacing the Wuhan strain as a globally prevalent strain, but with no reduction in vaccine protective effect compared to the Wuhan strain (Weissman et al., 2020); NA, not available.

The role of PBNA in the evaluation of therapeutic antibodies

PBNA has also been used to assess the in vitro efficacy of therapeutic monoclonal antibodies against WT SARS-CoV-2 and variants. The pseudovirus allowed the screening of antibodies with neutralizing effects and evaluated the antibodies’ effectiveness and their sites of action. PBNA analysis showed that the nanobody C5G2 synthesized by Zhao et al. (2022) could protect cells from infection by WT SARS-CoV-2 and VOCs, except for the Delta variant. LS-BB2z, a multivalent nanobody constructed by lumazine protein nanobody binding platform and monovalent nanobody, has enhanced the monovalent nanobody’s ability to neutralize the pseudovirus (Lu et al., 2023). A genetically encoded photosensitizer SOPP3 enhanced the neutralization efficacy of antibodies against S proteins and was effective against WT SARS-CoV-2, Delta, and Omicron variants (Yao et al., 2023). In addition, PBNA has also been used to locate specific mutations of variants that affect the neutralizing effect of antibodies to provide insight into the mechanisms of antibody resistance. In Wu et al.’s (2021b) study, 12 single deconvolution mutants based on the variations and the top five epidemic SARS-CoV-2 variants in the UK were tested using PBNA to determine the relative neutralization titers. They discovered that the N501Y, N439K, and S477N mutations dramatically reduced specific monoclonal antibodies’ ability to neutralize but had no appreciable impact on the power of convalescent sera and sera-induced by vaccination to neutralize. Wang et al. (2021b) established to evaluate all single mutant variations of the B.1.1.7 and B.1.351 mutants based on VSV-based SARS-CoV-2 pseudoviruses, a total of 18 mutant pseudoviruses to efficacy their effects on monoclonal antibody therapy and vaccine efficacy. It is shown that most monoclonal antibodies directed against the spike protein’s N-terminal domain (NTD) are ineffective against B.1.1.7. The B.1.351 mutant was found to be resistant to monoclonal antibodies against the spike protein NTD, and RDB due to the E484K substitution mutation (Wang et al., 2021b, p. 7).

Application of sars-cov-2 pseudovirus in drug screening and development

The infection of SARS-CoV-2 to host cells is related to the membranes fusion proces. This process involves binding S protein to cell membrane receptor ACE2 and cleavage of S protein by protease on cell membrane. According to this principle, therapeutic drugs are designed. The SARS-CoV-2 entry inhibitor blocks the pseudovirus from entering the cell, lowering the fluorescent protein production. It is the most suitable tool for drug development and screening at the cellular level.

Patil et al. (2023) constructed HIV-1-based SARS-CoV-2 pseudovirus to evaluate that Lactoferin hydrolysed for 90 min inhibits on CTSL and prevents the pseudovirus from entering target cells. [Pyr1]-Apelin-13 can bind to ACE2, inhibit the binding of S protein and cell ACE2, and effectively reduce the pseudovirus infection to 16HBE14o cells (Park et al., 2022). A flavonoid extract from Lophatherum gracile methanol extract and flavone C-glycoside isoorientin and PEDOT also inhibit the infection of HEK293T cells with hACE2 overexpression by WT and mutant pseudovirus by blocking the binding between SARS-CoV-2 and ACE2 (Hung et al., 2022; Chen et al., 2023a). Hesperidin (HD) and hesperitin (HT) are potential inhibitors of TMPRSS2, which can reduce the expression of ACE2 and TMPRSS2 and block the interaction between S protein and ACE2, and show antiviral activity against D614G and 501Y. V2 mutant pseudoviruses (Cheng et al., 2021). Heparan sulfate proteoglycan (HSPG) is necessary to enhance the binding of S protein to cell surface ACE2 and promote SARS-CoV-2 infection of host cells (Clausen et al., 2020). Dwivedi et al. (2022) can competitively inhibit the binding of HSPG to S protein by using marine sulfed-glycans, and reduce the infectivity of SARS-CoV-2 pseudovirus.

In the face of a global pandemic of the magnitude of COVID-19, a drug screening of existing drugs is necessary. Wang et al. (2021a) and Ge et al. (2021) used a pseudovirus drug screening system to investigate the antiviral impact of drugs. They concluded that astemizole and doxepin represent potential drug candidates that can be reused in anti-SARS-CoV-2 therapies (Ge et al., 2021; Wang et al., 2021a). Using successfully generated VSV pseudoviruses bearing the S protein, Xiong et al. (2020) evaluated the antiviral efficacy of 1,403 FDA-approved drugs against ACE2-expressing human BHK21 cells. Through three rounds of screening tests, five drugs that specifically inhibited pseudoviruses in vitro with >85% inhibition rate and no cytotoxicity were selected. They found that combination therapy could reduce the concentration of drugs to exert antiviral effects, thereby reducing the cytotoxic side effects, suggesting a new combination therapy idea for COVID-19 clinical treatment (Xiong et al., 2020). Meng et al. (2023) constructed an HIV-based SARS-CoV-2 pseudovirus that screened out 24 flavonoids that blocked the invasion of the pseudovirus into host cells. Hashizume et al. (2023) successfully screened 24 SARS-CoV-2 entry inhibitors from a US Food and Drug Administration-approved drug library. They determined that the main targets of these 24 compounds were dopamine receptor D2 antagonists, which led to the discovery of the broad-spectrum anti-SARS-CoV-2 activity of phenothiazines (Hashizume et al., 2023).

Conclusions

Of course, there are limitations to pseudovirus technology. First, the envelope glycoprotein distribution and conformation of pseudoviruses may differ slightly from the live virus, and it is vital to increase the diversity of various packaging systems. Second, PBNA cannot be used to quantify sera with low neutralizing antibody titers as the minimum serum dilution for PBNA is 1:25 (Hvidt et al., 2022). Furthermore, there is a risk of reverting to the live virus (Sakuma et al., 2010; Bilska, Tang & Montefiori, 2017). So, live virus detection results are still the gold standard. However, it is undeniable that SARS-CoV-2 pseudovirus have been shown to correlate well with live viruses in some areas, and their application has lowered the threshold of experimental techniques and greatly facilitated the study of SARS-CoV-2. We expect the development of SARS-CoV-2 pseudovirus applications to inspire more enveloped virus drugs and vaccines for highly infectious enveloped viruses, such as Ebola (Sokolova et al., 2021) and hantavirus (Mayor et al., 2021), as well as for emerging infectious diseases, such as monkeypox virus, which appeared in 2022 (Feng et al., 2022).

The authors are very thankful to everyone who participated in this study.

Additional Information and Declarations

Competing Interests

Author Contributions

Data Availability

The authors declare that they have no competing interests.

Conglian Tan conceived and designed the experiments, performed the experiments, analyzed the data, prepared figures and/or tables, authored or reviewed drafts of the article, and approved the final draft.

Nian Wang performed the experiments, authored or reviewed drafts of the article, and approved the final draft.

Shanshan Deng analyzed the data, prepared figures and/or tables, and approved the final draft.

Xiaoheng Wu analyzed the data, prepared figures and/or tables, and approved the final draft.

Changwu Yue analyzed the data, authored or reviewed drafts of the article, and approved the final draft.

Xu Jia conceived and designed the experiments, authored or reviewed drafts of the article, and approved the final draft.

Yuhong Lyu conceived and designed the experiments, authored or reviewed drafts of the article, and approved the final draft.

The following information was supplied regarding data availability:

This article is a review and no experiments were conducted, so no raw data or code are available.

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
