# Peer review of "The development and application of pseudoviruses: assessment of SARS-CoV-2 pseudoviruses"

_PeerJ, doi:10.7717/peerj.16234_

## Round 0.1 · original submission · Major Revisions

Our reviewers have found the need for important improvements. Please be sure to mention the other recent reviews of the topic and (most importantly) highlight how your review differs from the extant works and how it provides added value to the community.

Reviewer 1 ·

Basic reporting

I have identified several errors in wording and basic concepts throughout the manuscript. These issues suggest that the authors did not pay close attention to the clarity and accuracy of their writing, and as a result, the manuscript does not meet the standards required for publication in PeerJ.

For example (only a few of many points that are not carefully written)

In the abstract, please note that COVID-19 patients should be referred to as such, rather than as SARS-CoV-2 patients, as SARS-CoV-2 is the name of the virus, not the disease

I would suggest using 'coronavirus' instead of 'Corona Virus' in the manuscript, as 'coronavirus' is the generally accepted term for this family of viruses.

Line 44-46: I found the authors' explanation of the function of the spike protein to be confusing. I would suggest that they revise this section to make it more clear and concise.

Line 55: The authors mentioned SARS-CoV-2 pseudovirus but cited the paper published in 2002, which is 20 years before the virus was isolated

The authors did not carefully pay attention to the detail such as the abbreviation of Receptor-Binding Domain (RBD) in line 46

Overall, I have identified numerous issues with the manuscript, which indicate that significant refinement is required before it can be considered for publication. I would suggest that the authors carefully review their work and make the necessary revisions to address the issues I have raised

Experimental design

This manuscript aims to review the use of pseudovirus technology as a model for SARS-CoV-2. However, it lacks a specific study design. The reviewer expects the manuscript to provide up-to-date information on the technology's recent developments and how they can be employed to resolve current research issues. Although there have been similar reviews published, this manuscript is expected to present new information rather than repeat old work. Upon reading the manuscript, I have noticed that the authors have attempted to add some new information, but the details are not adequately elaborated. The language used is unclear, making it challenging for readers to understand the authors' intended message. More work is necessary to complete this review, and the current version is not suitable for publication.

Validity of the findings

Not to the standard of publication as it currently stands.

Additional comments

-

·

Basic reporting

no comment

Experimental design

no comment

Validity of the findings

no comment

Additional comments

This review summarizes the construction of SARS-CoV-2 pseudoviruses based on different packaging systems, current applications and the potential advantages and disadvantages of SARS-CoV-2 pseudovirus technology. I have some questions and suggestions.

(1)Chen et al. have already published a review titled “ Construction and applications of SARS-CoV-2 pseudoviruses: a mini review” (Int. J. Biol. Sci., 2021; 17(6): 1574-1580.) to discuss the construction of SARS-CoV-2 pseudoviruses based on different packaging systems, current applications, limitations, and further explorations. The authors should focus on the currently updated information about the SARS-CoV-2 pseudoviruses.
(2)There are some other replication-deficient SARS-CoV-2 pseudoviruses (PLoS Pathog., 2021; 17(3):e1009439; Sci. China Life Sci., 2022; 65(9):1894-1897.), which also should be discussed.
(3)The conclusions part of the manuscript can be divided into two parts, one is about the limitations and weakness of the SARS-CoV-2 pseudoviruses, another is about the further perspectives of the SARS-CoV-2 pseudoviruses.
(4)There are a lot of written mistakes in the manuscript, such as “Without the S protein, the virus cannot connect to the host cell's receptor or begin infection. The virus cannot infect a host cell without the S protein attached to the receptor” (page 7, lines 44-46). The abbreviation of the Receptor-Binding domain should be RBD (page 7, line 46). The (see Figure 1) should be in the front of “.”. “The Spike protein on the surface of SARS-CoV-2 pseudovirus has a high structural similarity to the SARS-CoV-2 envelope protein. SARS-CoV-2 pseudovirus surface proteins share a high degree of structural similarity with SARS-CoV-2 Spike protein.” (page 7, lines 56-58). In the conclusions part, line 283 “t the”. The authors need carefully check the whole manuscript.
(5)The authors need to describe the information about the used packaging systems in “1.2 Construction of relatively high titer SARS-CoV-2 pseudovirus” part.
(6)In Figure 2, the label “Finger 2” is not correct. In addition, the different genes incorporated into the vectors should be labeled using different colors to discriminated.

---

## Round 0.2 · Minor Revisions

I have taken over this manuscript from the previous handling editor. As such, this is the first time I am seeing the manuscript. I think your manuscript is fine in principle, but I would encourage you to do one round of critical editing and proof reading. In particular, the current text makes excessive use of the term "SARS-CoV-2" in my opinion. I realize that your paper is about SARS-CoV-2, but in some parts of the manuscript it seems every third word is SARS-CoV-2. If you can reword your text so this term occurs a bit less frequently that would be good.

I also disagree with your overall motivation in the first sentence of the abstract ("Although most Coronavirus disease (COVID-19) patients currently can recover fully, a proportion of the population still suffers from long-term adverse effects associated with SARS-CoV-2.") and similarly in lines 37-38 of the introduction. The contrast you set up "although most patients recover" is not useful in my opinion. Yes, there are long-term adverse effects, and they are important, but there are also still people who die from primary infection, and those cases are also important. Something like the following would be more accurate: "Although most COVID-19 patients can recover fully, the disease remains an important cause of morbidity and mortality. And in addition to the consequences of acute infection, a proportion of the population experiences long-term adverse effects associated with SARS-CoV-2."

·

Basic reporting

no comment

Experimental design

no comment

Validity of the findings

no comment

Additional comments

The authors have made substantial improvements to the manuscript, which can be accepted for publication.

---

## Round 0.3 · accepted · Accept

Thank you for your careful revisions.